# Regulation of therapeutic protein release in response to circadian biomarkers

Nik Franko [1], Shichao Li[2], Silvia Galvan [1], Zsoka Csorba[1], Ana Palma Teixeira[1], Mingqi Xie [2] & Martin Fussenegger [1,3] ✉

The human circadian clock integrates external environmental changes and internal physiological signals to generate natural oscillations of secreted endocrine signals to regulate diverse biological processes. Here, we explore human receptors responsive to molecules displaying in vivo oscillatory patterns and identify melatonin receptor 1A (MTNR1A) as a promising molecular sensor to trigger transgene expression. We engineer a melatonin-inducible gene switch consisting of ectopically expressed MTNR1A linked to an amplifier module utilizing the native $G_{\alpha s}$ protein-mediated cell signaling cascade, which involves adenylyl cyclase, cAMP, protein kinase A and the cAMP-responsive transcription factor CREB, to drive transgene expression from a synthetic promoter. This system operates within the physiological melatonin concentration range, selectively responding to night-phase levels of the diurnal rhythm, while remaining unresponsive to day-phase levels. Such temporal control suggests its potential for personalized cell- and gene-based therapies requiring once-per-day dosing regimen. As proof-of-concept, we show that alginate-encapsulated engineered cells implanted in C3H/HeJ male mice can translate circadian inputs or clinically licensed MTNR1A agonists into regulated GLP-1 expression as a therapeutic output exclusively secreted during nighttime, highlighting potential as an experimental cell therapy for obesity-dependent type-2 diabetes.

In recent years, significant progress has been made in engineering cells to respond in a programmable, user-defined manner. Synthetic biology[1] approaches have enabled engineered cells to exhibit complex computer-like behavior[2–4], endowing them with the ability to conduct diverse, non-native tasks[5–7]. Control over engineered biological systems, reliant on specific inputs, can be achieved at the transcriptional[8–10], translational[11,12] or post-translational level[13–17], with transcriptionally controlled systems being the most widely adopted due to their robustness[18,19]. Many synthetic transcriptional switches are based on bacterial transcription factors responsive to small-molecule inputs, fused to viral or mammalian transactivation domains to drive transgene expression in mammalian cells[9,10,20]. Although these have been adopted to control expression of various therapeutic genes in

mouse studies, they often require high concentrations of the input molecules, making them unsuitable for long-term translational applications. Alternative approaches to engineer sense-response systems leverage mammalian cell membrane receptors[21], channels[22], or intracellular sensing components[23]. These components are less immunogenic and are more likely to detect physiological or disease-relevant input concentrations. The signal transmission of these receptors often depends on native cell signaling cascades, which culminate in the activation of transgene expression from a synthetic promoter. Taking advantage of a vast repertoire of these mammalian sensors, cells have been engineered to sense microbe-derived formyl peptides[24], bile acids[25], menthol[21], fatty acids[26], inflammatory cytokines[27,28], and to activate therapeutic gene expression in mouse models of liver injury[25],

[1]Department of Biosystems Science and Engineering, ETH Zurich, Basel, Switzerland. [2]Westlake Laboratory of Life Sciences and Biomedicine, Hangzhou, Zhejiang, China. [3]Faculty of Science, University of Basel, Basel, Switzerland. ✉e-mail: fussenegger@bsse.ethz.ch

chronic pain[29], and psoriasis[27]. Both closed- and open-loop controlled biological systems offer promising approaches for precise regulation of cell-based therapies in response to physiologically relevant cues.

Many organisms, including humans, exhibit natural oscillations that affect a range of biological processes. These rhythmic fluctuations, known as circadian rhythms, are governed by an autonomous, intrinsic timekeeping system called the circadian clock. This clock operates in a roughly 24-h cycle and integrates external environmental changes and internal physiological cues to enable robust adaptation to the surrounding environment[30,31]. Environmental cues, termed zeitgebers, entrain the circadian rhythm by either advancing or delaying the circadian clock, thus ensuring synchronization with the solar day. The primary circadian clock, located in the suprachiasmatic nucleus of the hypothalamus coordinates various brain areas and peripheral tissues throughout the body via neural and hormonal signals[32]. It regulates the circadian secretion of diffusible endocrine signals, such as thyrotropin (TSH)[33], melatonin (MTN)[34], and cortisol[35], which exhibit an oscillatory pattern over the course of the day. We hypothesized that such endogenous rhythmic signals could be repurposed to power the expression of therapeutic genes in engineered cells, enabling circadian regulation of cell therapies. As cortisol levels are highly susceptible to stress and external perturbations[36,37], we focused on TSH and MTN as more reliable candidates for circadian input signals.

To implement this strategy, we design a gene switch using human receptors responsive to these circadian hormones and their associated signaling cascades. After screening various candidates, we identify the melatonin receptor 1A (MTNR1A), which relies on the native cAMP signaling cascade for signal transmission, as the most promising component for constructing a circadian gene switch. Fine-tuning MTNR1A expression and optimizing the reporter construct yielded a system with minimal leakiness and a high dynamic range, responsive specifically to melatonin levels characteristic of the nocturnal phase, while remaining inactive during daytime levels. The switch exhibits tunable, robust, and reversible kinetics of transgene expression. To illustrate the therapeutic potential of this circadian rhythm sense-response system, we connect the melatonin-sensing module to a glucagon-like peptide-1 (GLP-1) expression actuator module. GLP-1 is a clinically approved treatment for type-2 diabetes and obesity. Engineered cells implanted in mice detect and respond to both physiological and experimentally manipulated

melatonin, producing GLP-1 at night or in response to exogenous melatonin. In type-2 diabetic mice, implantation of these engineered cells successfully restores normoglycemia through melatonin-induced GLP-1 release. This work presents a proof-of-concept for harnessing endogenous hormonal rhythms to drive therapeutic gene expression in engineered cells. Such circadian gene switches hold promise for a wide range of applications, including drug discovery, chronobiology research and precise regulation of next-generation cell and gene therapies.

## Results

### Design of mammalian circadian rhythm sense-response systems

To build a circadian rhythm-controlled transcription system, we focused on two circadian biomarkers that display pulsatile levels in the bloodstream, TSH and melatonin, which are both present at low levels during the day and peak during the night (Fig. 1a). An ideal circadian sense-response system would exhibit minimal background transgene expression under day-time hormone levels, while elevated night-time levels would trigger receptor activation, and robust transgene expression (Fig. 1b). We screened candidate human G protein-coupled receptors (GPCRs) that could serve as biosensors for these inputs and translate a circadian signal to transcription activation of desired transgenes. Specifically, we tested the TSH receptor (TSHR) and the melatonin receptors MTNR1A and MTNR1B. All candidate systems were evaluated in transiently transfected human embryonic kidney (HEK293T) cells using human placental secreted embryonic alkaline phosphatase (SEAP)[38] as a reporter protein, expressed under the control of a synthetic promoter containing cAMP response elements (CRE) to monitor activation of the cAMP signaling pathway. TSH stimulation led to SEAP expression via activation of constitutively expressed TSHR (Fig. 1c); however, high basal leakiness in the absence of TSH resulted in poor inducibility. In contrast, of the two melatonin receptors tested, MTNR1A showed strong performance with low background activity and robust SEAP induction upon melatonin stimulation (Fig. 1d). MTNR1B, on the other hand, failed to elicit melatonin-dependent SEAP expression (Fig. 1d). Based on these results, the melatonin-inducible system using MTNR1A was identified as the most promising approach for circadian rhythm-driven transgene expression and was selected for further optimization and characterization.

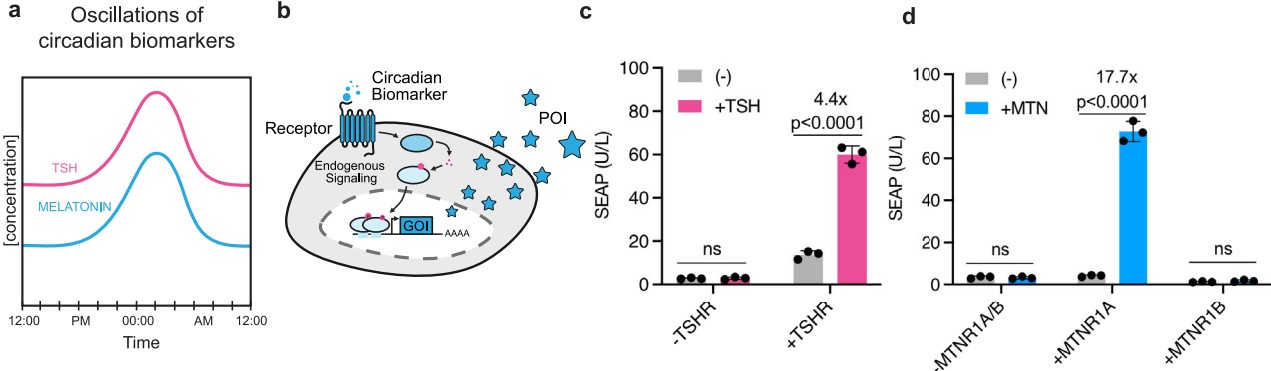

**Fig. 1 | Design of mammalian circadian rhythm sense-response systems.**
**a** Oscillatory pattern of circadian biomarkers TSH and melatonin in vivo, showing distinct peaks during the night. **b** Schematics of a cell-based system responsive to circadian biomarker. Circadian rhythm receptor ectopically expressed on the cell surface monitors circadian biomarker levels. Elevated concentration triggers the receptor and activates a signaling pathway that subsequently leads to transgene expression activation and secretion of protein of interest (POI) from the cells. **c** Transgene expression activation by GPCR-based TSHR receptor. Cells were transfected with TSHR ($P_{SV40}$-TSHR-pA, pSG14) and a reporter construct containing a $P_{cAMP}$-driven SEAP (pVH421) expression cassette. SEAP secretion levels were

determined 24 h after induction with TSH (30 pM). **d** Transgene expression activation by GPCR-based melatonin receptors: MTNR1A and MTNR1B. HEK293T cells were co-transfected with a combination of melatonin receptor MTNR1A ($P_{mPGK}$-MTNR1A-pA, pNF111) or MTNR1B ($P_{CMV}$-MTNR1B-pA, pNF82) and a reporter construct containing $P_{cAMP}$-driven SEAP expression cassette (pNF421). After induction with 100 nM melatonin (MTN) for 24 h, SEAP secretion levels in the cell supernatant were determined. In (**c**, **d**), data are shown as mean ± SD of $n$ = 3 biological replicates. Individual data are shown as filled circles with fold induction indicated above the bars. In (**c**, **d**), statistical significance was calculated with a two-tailed unpaired $t$-test. Source data are provided as a Source Data file.

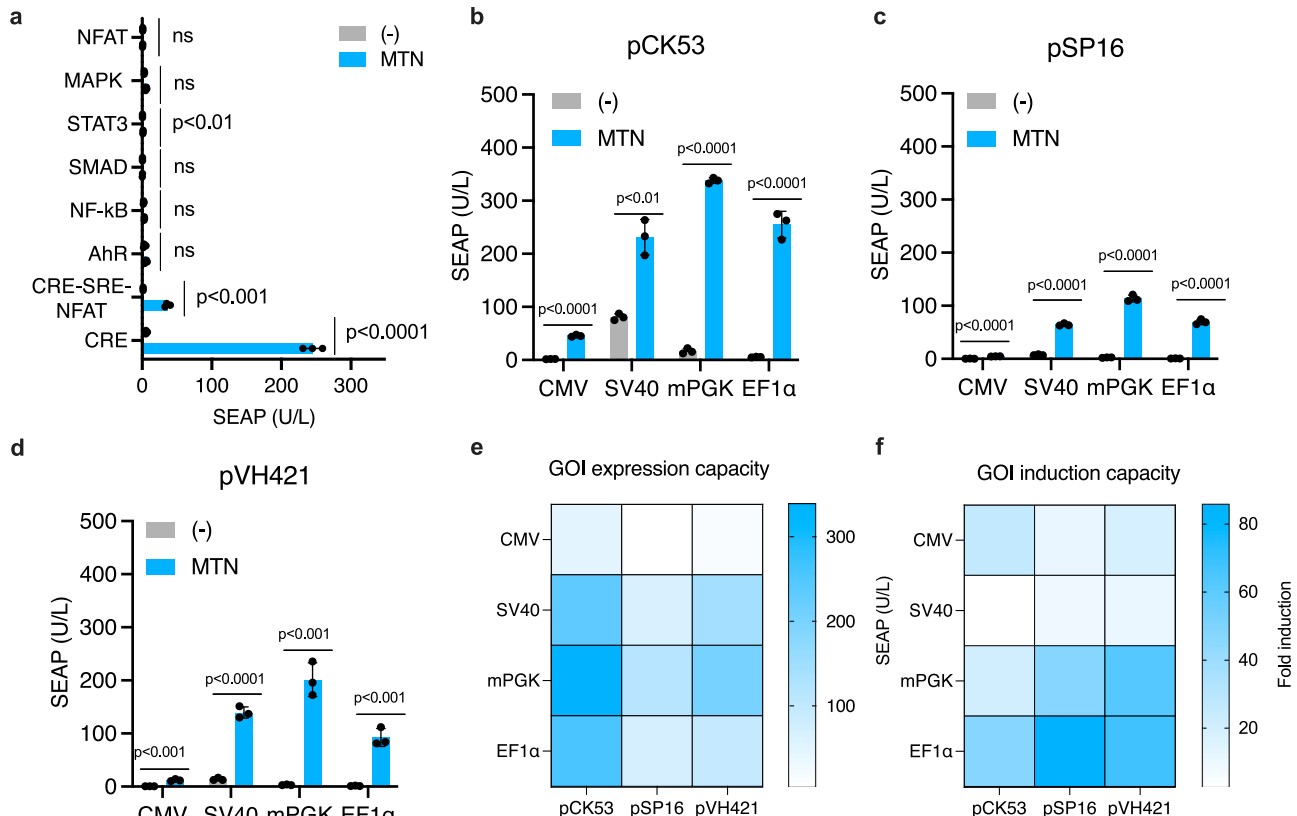

**Fig. 2 | Optimization of melatonin sense-response system. a** MTNR1A signaling orthogonality. MTNR1A was co-transfected with reporters for the indicated signaling pathways, followed by MTN induction for 24 h before determining SEAP expression. **b–d** Effects of various constitutive promoters driving MTNR1A expression and $P_{cAMP}$ promoter variants on the system's inducibility and SEAP expression capacity. MTNR1A was placed under the control of four constitutive promoters with different strengths, namely $P_{SV40}$, $P_{CMV}$, $P_{mPGK}$, and $P_{EF1\alpha}$. After combinatorial co-transfection with $P_{cAMP}$ promoter variants (pCK53, pSP16, and pVH421; see Supplementary Table 1 for details) driving SEAP expression, cells were treated with melatonin (100 nM) and SEAP was profiled 24 h thereafter. **e, f** Heat-maps illustrating the performance of promoter and reporter combinations presented as maximal transgene expression capacity (**e**) and fold-inductions, calculated as a ratio of melatonin-induced to un-induced SEAP expression levels (**f**) ($n = 3$ biological replicates). In (**a–d**), data are shown as mean ± SD of $n = 3$ biological replicates, with individual data shown as filled circles. Statistical significance was calculated with a two-tailed unpaired $t$-test. Source data are provided as a Source Data file.

## Tuning the melatonin-responsive transcriptional switch

Given that several pathways have been implicated in melatonin signaling[39,40], we assessed whether MTNR1A activation by melatonin could induce transcription from a range of reporter promoters responsive to calcium signaling (NFAT), mitogen-activated protein kinase (MAPK/ERK), JAK/STAT, NF-κB, TGF-β/Smad and AhR. Using transient co-transfection assays, we observed that melatonin-inducible transgene expression was restricted to constructs containing CRE sites (Fig. 2a), confirming that in our system, MTNR1A primarily signals through the cAMP pathway. To increase the performance of the circadian rhythm-responsive gene expression system, we systematically screened combinations of constitutive promoters driving MTNR1A expression (CMV, SV40, mPGK, or EF1α) with different synthetic CRE-containing promoters controlling SEAP expression, in order to identify configurations offering both high inducibility and strong transgene expression capacity. We evaluated three reporter plasmids, pCK53 (Fig. 2b), pSP16 (Fig. 2c), and pVH421 (Fig. 2d), which differ in the spacer sequences between CRE sites and their downstream minimal promoters (Supplementary Table 1). Head-to-head comparisons identified the combination of the mPGK promoter ($P_{mPGK}$) driving MTNR1A and the pVH421 reporter as the most effective, yielding the highest SEAP expression and robust fold induction in response to melatonin (Fig. 2e, f). This optimized configuration was used in all further experiments.

## Characterization of the melatonin-responsive system

To assess the generalizability of the system, we tested its functionality across multiple mammalian cell lines, including Chinese hamster ovary cells (CHO), widely used in biopharmaceutical manufacturing, and human mesenchymal stem cells (hMSC), a common cell chassis in many cell therapy trials[7]. Broad responsiveness to melatonin was observed across all tested lines, although to varying extents, likely reflecting differences in transfection efficiency (Fig. 3a). In humans, plasma melatonin levels typically oscillate from low pM levels during the day phase[41] to nighttime peaks of up to 700 pM[42]. We therefore evaluated whether our transcriptional switch exhibits appropriate sensitivity to the varying melatonin levels to differentiate between circadian phases. Dose-response assays showed that SEAP expression was significantly induced at melatonin concentrations as low as 100 pM (Fig. 3b), aligning with night-time physiological levels.

To explore the potential for open-loop control, we tested the system's responsiveness to four clinically approved MTNR1A agonists, namely piromelatine, tasimelteon, ramelteon, and agomelatine[43] (Fig. 3c). These compounds all triggered dose-dependent SEAP expression, offering a pharmacologically tunable method for controlling transgene output. Similar sensitivities were observed across all the agonists, except for piromelatine, which exhibited a lower dynamic range. Importantly, these agonists feature extended half-lives compared to melatonin, making them more suitable for sustained in vivo applications. Next, we characterized the expression kinetics of the

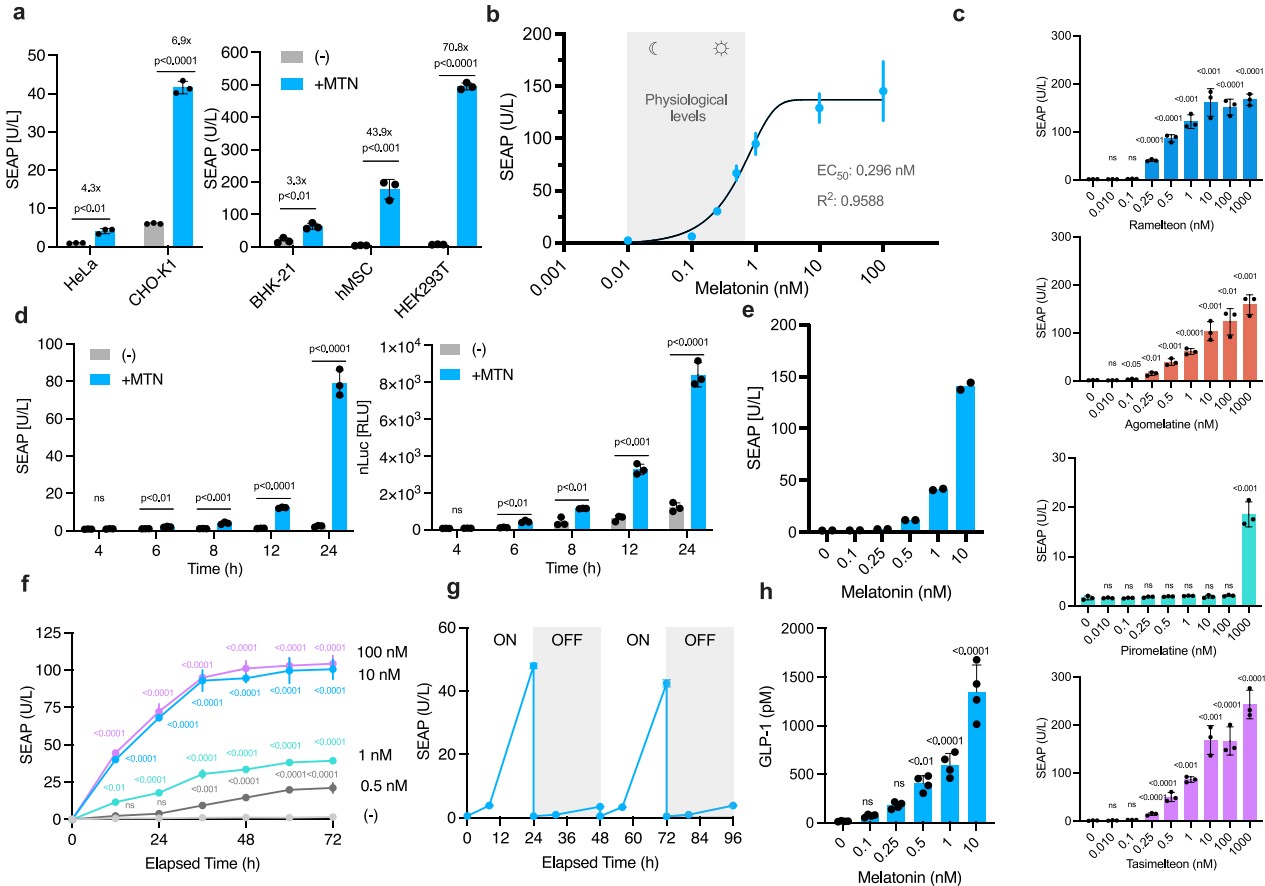

**Fig. 3 | Characterization of circadian rhythm sense-response system.**
**a** Melatonin-inducible transgene expression across various mammalian cell lines transfected with MTNR1A (pNF111) and $P_{cAMP}$ driven SEAP reporter (pVH421). SEAP activity was determined 24 h after MTN (100 nM) treatment. **b** Sensitivity and tunability of the circadian rhythm sense-response system in HEK293T cells co-transfected with pNF111 and pVH421. SEAP expression was profiled 24 h after addition of indicated melatonin levels. **c** Sensitivity and adjustability of the circadian rhythm sense-response system by clinically approved drugs. Cells co-transfected with pNF111 and pVH421 were induced with MTNR1A agonists at the indicated concentrations for 24 h before determining SEAP expression levels. **d** Melatonin-induced secretion kinetics of SEAP and nLuc from HEK293T cells transfected with pNF111 and either SEAP (pVH421) or nLuc (pNF392) reporter plasmids. Cells were induced with 10 nM MTN and SEAP or nLuc expression levels were determined at indicated time points after induction. **e** Melatonin-dependent SEAP secretion from a monoclonal cell line (C16) with a genome integrated MTNR1A and $P_{cAMP}$ driven SEAP reporter construct. SEAP activity was profiled 24 h after induction with indicated MTN concentrations. **f** SEAP secretion dynamics from clone C16. Cells were treated with specified MTN concentrations and SEAP levels were determined at indicated time points. **g** Reversibility of melatonin-induced SEAP expression. Cells were cultivated for 96 h, with media exchanged every 24 h, alternating between standard or melatonin (10 nM)-supplemented medium. SEAP levels were determined at indicated time points. **h** Melatonin-dependent GLP-1 secretion from a monoclonal cell line (C33) with a genome integrated MTNR1A (pNF396) and $P_{cAMP}$ driven GLP-1 construct (pNF395). GLP-1 levels were determined by ELISA 48 h after induction with the indicated MTN concentrations. In (**a–d**, **f**, **g**), data are shown as mean ± SD of $n$ = 3 biological replicates and in (**e**) as mean of $n$ = 2 biological replicates. In (**h**), data are shown as mean ± SD of $n$ = 4 biological replicates. Where no bar is shown, the SD is smaller than the symbol. In (**a**, **c**, **d**, **h**), individual data are shown as filled circles. Statistical significance was calculated with a two-tailed unpaired $t$-test in (**a**, **d**) or one-way ANOVA in (**c**, **f**, **h**). Source data are provided as a Source Data file.

system by tracking the levels of SEAP and nanoluciferase (nLuc) over time, following melatonin stimulation (Fig. 3d). A significant increase in reporter expression was observed at 6 h post-induction in comparison to non-stimulated controls.

For long-term experiments, we established cell populations with genomically integrated switch components using a Sleeping Beauty transposase-based system[44]. Single cell clones were isolated via FACS, and several high-performing clones were identified based on transgene expression and fold induction (Supplementary Fig. 1a, b). One of the top performing clones showed up to 40-fold induction and exhibited dose-dependent SEAP expression upon melatonin treatment (Fig. 3e). We further profiled its secretion dynamics across a range of melatonin concentrations over time (Fig. 3f). Additionally, the system could be toggled ON or OFF by alternately culturing the cells in melatonin-containing or melatonin-free medium, demonstrating reversible transgene expression (Fig. 3g).

Next, to showcase the therapeutic ability of the circadian rhythm sense-response system, we focused on obesity, a major health burden in the 21st century[45,46], and selected GLP-1[47] as the effector molecule due to its well-established clinical efficacy[47]. We connected the melatonin-sensing module to an actuator encoding GLP-1 and nLuc, separated by a self-cleaving peptide, allowing simultaneous regulation and streamlined screening. A stable cell population with genomically integrated switch components was generated, and high-performing clones were identified based on nLuc expression and fold induction (Supplementary Fig. 2a). Secondary screening of the top five candidates confirmed dose-dependent production of the transgene products (Supplementary Fig. 2b), with clone 33 ($C_{33}$) identified as the top performer; this clone was selected for all further experiments. This optimized cell line, named HEK$_{GLP-1}$, exhibited melatonin-dependent GLP-1 production, with an induction level of above 100 pM (Fig. 3h)—aligning with physiological night-time melatonin levels.

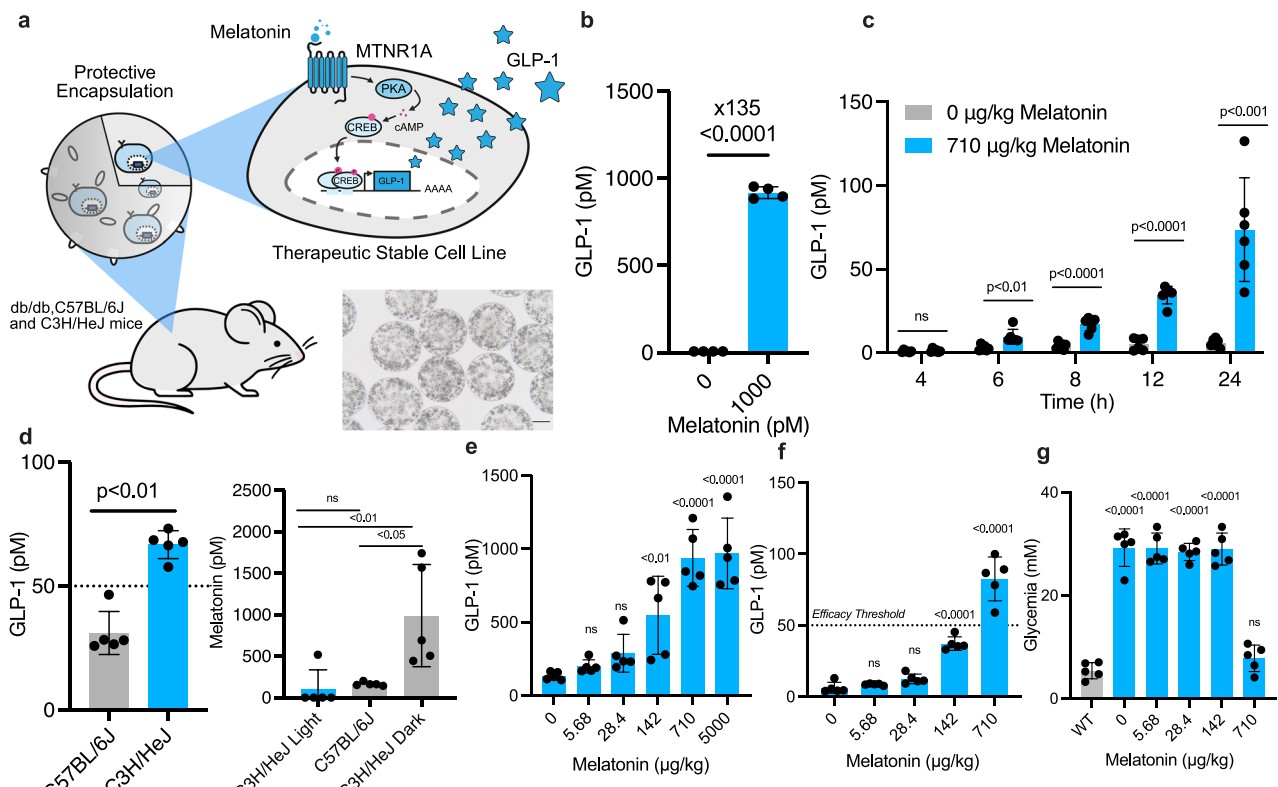

**Fig. 4 | Validation of the melatonin-induced therapeutic transgene product expression in mice. a** Schematics of HEK$_{GLP-1}$ cells, microencapsulated in microbeads and implanted into mice. The synthetic gene network continuously tracks the phase of the diurnal rhythm by sensing environmental melatonin levels and programs cells to produce GLP-1 when melatonin levels are high. The microscopy image shows cells inside alginate-poly-L-lysine-alginate beads (scale bar: 200 μm). **b** GLP-1 secretion from encapsulated HEK$_{GLP-1}$ cells cultivated without or with melatonin (1 nM) profiled in supernatants 48 h after induction. Numbers show fold-changes between GLP-1 levels of melatonin-treated and GLP-1 levels of DMSO-treated (control) samples. **c** Activation kinetics of melatonin-dependent target gene expression in vivo. One day after implantation of $5 \times 10^6$ microencapsulated HEK$_{GLP-1}$ cells into C57BL/6J mice, animals were given a single oral dose of 710 μg/kg melatonin, then GLP-1 levels in the bloodstream were monitored over 24 h. Mice receiving ddH$_2$O were used as negative controls. **d** GLP-1 activation in response to endogenous melatonin levels (left) and light-dependent manipulation of endogenous melatonin production (right) in mice. Male C3H/HeJ or C57BL/6J mice were housed in the dark or exposed to constant illumination (1 mW/cm$^2$, 24 h) before receiving intraperitoneal implants of microencapsulated HEK$_{GLP-1}$ ($5 \times 10^6$ cells per mouse). GLP-1 levels (right) and corresponding blood melatonin levels (left) were quantified 24 h after implantation. **e** Melatonin-dependent transgene expression in vivo. One day after implantation of microencapsulated HEK$_{GLP-1}$ cells into C57BL/6J mice, animals received various doses of melatonin and GLP-1 levels in the bloodstream were quantified after 24 h. **f, g** Therapeutic efficacy of melatonin-dependent GLP-1 production. db/db mice received intraperitoneal implants of microencapsulated HEK$_{GLP-1}$ ($5 \times 10^6$ cells per mouse) and indicated MTN oral dose. GLP-1 and fasting glycemia levels were recorded after 24 h. C57BL/6J mice (WT) were used as healthy controls. Data in (**b**) are shown as mean ± SD ($n$ = 4 biological replicates) and in (**c–g**) as mean ± SEM ($n$ = 5 mice per group). Therapeutically active GLP-1 concentrations in the bloodstream of db/db mice are indicated by a dashed line ( > 50 pM; efficacy threshold). Statistical significance was calculated with a two-tailed unpaired $t$-test in (**b–d**) or one-way ANOVA in (**e, f, g**). Source data are provided as a Source Data file.

## In vivo validation of melatonin-induced GLP-1 secretion

To evaluate the therapeutic potential of the melatonin-responsive system in vivo, HEK$_{GLP-1}$ cells were encapsulated in coherent semipermeable alginate-poly-L-lysine-alginate microbeads (Fig. 4a), a clinically available cell delivery strategy for treatment of diabetes[48]. In vitro, melatonin treatment of encapsulated cells led to robust GLP-1 secretion compared to untreated controls, confirming effective induction within the capsule matrix (Fig. 4b). Encapsulated HEK$_{GLP-1}$ cells were then implanted into melatonin-deficient C57BL/6J mice (Supplementary Fig. 3). Following oral administration of exogenous melatonin, GLP-1 secretion was induced in vivo (Fig. 4c), with similar activation kinetics to the reporter proteins in vitro (Fig. 3b). To evaluate the potential to synchronize therapeutic transgene expression with endogenous melatonin oscillations, we implanted encapsulated HEK$_{GLP-1}$ into C3H/HeJ mice[49], which produce melatonin endogenously (Supplementary Fig. 3). Circulating GLP-1 levels were significantly higher in C3H/HeJ mice compared to C57BL/6J mice, indicating activation of HEK$_{GLP-1}$ by endogenous melatonin and robust therapeutic gene expression (Fig. 4d, left). These results confirmed the existence of strain-specific differences in melatonin production. As expected,

melatonin levels increased under light deprivation, which stimulates pineal synthesis[50], and were suppressed under constant illumination (Fig. 4d, **right**), as determined by a custom-designed reporter assay (Supplementary Fig. 4). It is important to note, however, that mouse and human circadian physiology and melatonin profiles differ significantly[51,52]. Further studies will be required to determine precisely how therapeutic transgene expression can be coupled to the human circadian rhythm.

To explore pharmacologically controlled activation, we leveraged the melatonin-deficient background of C57BL/6J mice[49], where basal GLP-1 expression remained below the ~50 pM threshold required for therapeutic efficacy[53]. This low baseline effectively locked the system in an OFF state (Fig. 4d, **left**). Upon oral melatonin administration, GLP-1 expression increased in a dose-dependent manner (Fig. 4e), demonstrating controllability of the gene switch with this clinically approved over-the-counter drug typically taken at night. Finally, we evaluated therapeutic efficacy in type-2 diabetic db/db mice. Implantation of encapsulated HEK$_{GLP-1}$ cells followed by titrated oral melatonin dosing led to graded increases in circulating GLP-1 levels (Fig. 4f), ultimately restoring normoglycemia once therapeutically effective

thresholds were reached (Fig. 4g). This suggests the feasibility of developing personalized nighttime medications where therapeutic protein secretion is precisely coupled to endogenous or exogenous sleep-promoting hormones in the bloodstream.

## Discussion

The circadian rhythm sense-and-response system developed in this study expands the synthetic biology toolbox by enriching the repertoire of receptor-based gene-switches[54] available to augment human cells with new functionalities operating in a predictable, user-defined manner. To minimize immunogenicity and maximize translational potential, we focused on human-based sensors to build the gene switch. Among the receptors tested for their ability to sense molecules with oscillatory patterns in vivo, MTNR1A conferred the most robust transgene activation in response to elevated melatonin. While previous works report decreased cAMP levels upon MTNR1A activation[39], our results suggest that HEK293T cells overexpressing MTNR1A can elevate cAMP levels in response to melatonin, leading to CREB activation and induction of cAMP-dependent promoters. We confirmed that only promoters containing CRE were responsive to melatonin, while synthetic promoters driven by alternative signaling pathways were inactive.

We systematically characterized different combinations of constitutive promoters driving MTNR1A expression and synthetic promoters driving transgene expression, achieving either tight regulation of transgene expression or a balance of high expression and low leakiness, thereby making it possible to tailor system performance for specific applications. Importantly, the sensitivity of the developed system matches physiologically relevant melatonin concentrations[41], so that the system is switched OFF at daytime melatonin levels, while the increase of melatonin during the night-phase of the diurnal rhythm triggers robust transgene expression, effectively distinguishing between phases of the circadian rhythm cycle. However, it should be noted that endogenous melatonin levels decline with age[41], and this may cause insufficient therapeutic transgene activation in vivo, potentially limiting translational applications in older populations. Nevertheless, in such cases, readily available melatonin supplements[55] could be used to provide sufficient levels to activate transgene expression. Alternatively, we showed that clinically approved melatonin receptor drug agonists[43], such as Valdoxan® or Hetlioz®, with improved pharmacodynamic and pharmacokinetic properties, can trigger transgene expression, and thus could also enable exogenous control of the therapeutic output.

The present system with GLP-1 as a therapeutic output may have translational potential for circadian rhythm-regulated once-daily therapy for obesity[47] or type-2 diabetes treatment[56]. The engineered cells enabled melatonin-dependent GLP-1 expression, with robust GLP-1 production at melatonin levels that occur in the night phase of the diurnal rhythm. Further, when implanted in vivo into C3H/HeJ mice—a standard mouse model with an intact biosynthesis pathway for endogenous melatonin[49,52]—the cells could distinguish different melatonin concentrations produced by circadian manipulations, producing corresponding GLP-1 output levels. In contrast, upon implantation into melatonin-deficient C57BL/6J mice, we showed how externally applied melatonin enabled high and dose-dependent activation of GLP-1 release into the circulation. Furthermore, this strategy enabled restoration of normoglycemia in type-2 diabetic db/db mice. In providing a cell-therapy approach where therapeutic protein secretion is precisely coupled to circadian biomarkers, this work may pave the way to the development of personalized night-time medication regimens for patients suffering from sleep disorder-associated metabolic diseases. Current treatment regimens using GLP-1-derived medications, such as semaglutide[57], are typically administered as once-weekly injections, which can be inconvenient for long-term use from the patient's perspective and may cause undesirable side effects[58,59] due to high systemic doses. A promising alternative is gene therapy delivering constitutively expressed GLP-1 via viral vectors such as adeno-associated virus[60]. However, this approach lacks control over therapeutic protein levels once expression is established in the patient. Our system addresses this limitation by leveraging endogenous melatonin oscillations for basal control, while also permitting external modulation of transgene expression. MTNR1A agonists can be used to enhance GLP-1 production, while antagonists[61] could allow suppression in the case of adverse effects. This dual-mode regulation promises precise therapeutic dosing, minimizes side effects, and holds the potential for improved clinical outcomes.

We believe this system provides a foundation for gene- and cell-based therapeutic applications that require once-daily therapeutic dosing. Further, the actuator is modular and compatible with a wide range of transgenes beyond GLP-1, enabling easy substitution with alternative therapeutic proteins—such as monoclonal antibodies, erythropoietin[62], or other biologics to address diverse disease contexts. This would be particularly valuable for chronic conditions requiring steady-state therapeutics, while still allowing dose adjustment in the form of a pill. Alternatively, as a gene-based therapy, the genetic components of the circadian sense-response system could be delivered to cells exposed to higher melatonin levels, such as those close to the pineal gland or cerebrospinal fluid, where melatonin levels can be up to 20 times higher than in the general circulation[63].

The platform could also serve as a cell-based biosensor for studying circadian regulation in vivo. Engineered cells responsive to melatonin fluctuations could be used to investigate how specific biological or environmental conditions influence melatonin dynamics. A luminescence-based output would enable non-invasive, real-time monitoring in living organisms.

Due to its high sensitivity and robust performance, the system has strong potential for high-throughput screening of candidate small molecules or biologics for the ability to act as MTNR1A agonists, providing a high-performance, cell-based tool that should greatly facilitate the drug discovery process[43]. Its characteristics make it well-suited for identifying therapeutic candidates while providing deeper insights into physiological function—accelerating the discovery of new therapeutics targeting circadian regulation, sleep disorders, and related physiological processes. Finally, our system complements recent chronogenetic approaches that use core clock gene promoters[64], contributing to the advancement of the field of chrono-pharmacology[65]. By leveraging the temporal dynamics of hormonal rhythms, this platform opens up avenues for circadian-informed therapeutic interventions.

## Methods

### Ethical statement

This study was carried out in full compliance with all relevant ethical regulations and animal welfare legislation in China. The experiments were approved by the Institutional Animal Care and Use Committee (IACUC) of Westlake University, conducted in accordance with the Animal Care Guidelines of the Ministry of Science and Technology of the People's Republic of China, and performed by Shichao Li under the approved protocol (Protocol IDs: 20-009-XMQ and 25-013-XMQ). To reduce variability associated with hormonal fluctuations, only male mice were used. Mice were housed under a 12-h light–dark cycle, with five animals per cage. The ambient temperature was maintained at $21 \pm 1\,°C$, with humidity at $50 \pm 10\%$.

### Plasmid construction

The design and cloning details for all genetic constructs used in the study are provided in Supplementary Data 1. DNA constructs were generated by classic restriction and digestion molecular biology approach. To obtain the target DNA fragments, the polymerase chain reactions were performed using Q5 High-Fidelity DNA polymerase (M0491L, New England BioLabs) and purified by gel electrophoresis.

Backbone plasmids and target DNA fragments were digested with restriction enzymes (New England BioLabs), followed by ligation using T4 DNA ligase (EL0011, Thermo Fisher Scientific). After the ligation step, the plasmids were transformed and amplified in *Escherichia coli* strain XL10-Gold® (XL10-Gold® ultracompetent cells; 200314, Agilent Technologies) by overnight growth in Luria-Bertani (LB) broth. Next, the plasmid DNA was obtained by extraction from bacterial cells using ZR Plasmid Miniprep−Classic kit (D4054, Zymo Research). All steps were done according to the manufacturer's instructions. DNA constructs were verified by Sanger sequencing (Microsynth AG).

## Cell culture

Human embryonic kidney cells (HEK293T, ATCC: CRL-11268), human cervical adenocarcinoma cells (HeLa, ATCC: CCL-2), Chinese hamster ovary cells (CHO-K1, ATCC: CCL-61), human telomerase-immortalized mesenchymal stem cells (hMSC-TERT), baby hamster kidney cells (BHK-21, ATCC: CCL-10) were cultured in Dulbecco's modified Eagle's medium (DMEM; 61965026, Thermo Fisher Scientific) supplemented with 10% (v/v) fetal bovine serum (FBS; F7524, Sigma-Aldrich) in a humidified environment containing 7.5% $CO_2$ at 37 °C. Cells were passaged regularly every 2–4 days, depending on the cell density observed under the microscope or the pH of the culture media. After incubation in 2 mL of 0.05% trypsin-EDTA (25300054, Thermo Fisher Scientific) for 5 min at room temperature, cells were transferred into 8 mL of culture medium, centrifuged for 1 min at $200 \times g$, resuspended in fresh medium and re-seeded into a new plate for continuous culture or seeded into a specific plate at the desired cell density, depending on the experimental setup.

## Transient transfection of mammalian cells

For transient transfection experiments with pDNA, cells were seeded in transparent 96-well plates (3599, Corning), at $2 \times 10^4$ cells per well, 24 h before transfection. Cells in each well were transfected with a mixture containing 120 ng plasmid DNA and 600 ng polyethyleneimine (MW 40,000; 24765, Polysciences) in 50 μL DMEM without FBS. The mixture was incubated at room temperature for 20 min before being added dropwise to the cells. The cells were then incubated overnight. The following day, the transfection mixture containing medium was replaced with fresh DMEM supplemented with FBS, and with or without inducer, depending on the experimental condition. Reporter protein expression was assessed 24 h later, unless noted otherwise.

## Generation of monoclonal stable cell lines

Polyclonal stable cell lines were generated by co-transfecting HEK293T cells with a Sleeping Beauty (SB) transposase (pTS395) expression vector in a 1:5:5 (w/w) ratio with pNF394 and pNF396 or pNF395 and pNF396, containing SB recognition sites and encoding either a puromycin resistance marker and YPet fluorescent protein or blasticidin resistance marker and iRFP fluorescent protein. The medium was exchanged 12 h after transfection and cells were incubated for 48 h before the addition of selection medium containing puromycin and blasticidin. After five passages, the surviving polyclonal population was sorted by fluorescence-activated cell sorting (FACS) based on expression of fluorescent markers, YPet (517/530 nm) and iRFP (690/713 nm) (Supplementary Fig. 5). Cells showing double-positive fluorescence signal were sorted as single clones into a 96-well plate format to obtain 35 (cells engineered with pNF394 and pNF396) or 60 (cells engineered with pNF395 and pNF396) monoclonal cell lines. Single clones were expanded for two weeks before they were tested for melatonin responsiveness.

## Inducer preparation

TSH (T8931, Sigma) was prepared as a 1mM solution. Melatonin (HY-B0075, MedChemExpress) was prepared as 10 mM solution in DMSO. Ramelteon (HY-A0014, MedChemExpress), piromelatine (HY-105285,

MedChemExpress), agomelatine (HY-17038, MedChemExpress), and tasimelteon (HY-14803, MedChemExpress) were supplied as 10 mM solutions in DMSO. Working solutions were prepared by serially diluting the stock solutions in FBS containing DMEM.

## SEAP quantification

SEAP levels in cell culture supernatants were quantified by colorimetric assay that measures the increase of absorbance due to hydrolysis of para-nitrophenyl phosphate (pNPP) by SEAP protein. Samples were prepared by heat-inactivating (30 min, 65 °C) cell culture supernatant. After inactivation, 40 μL cell culture supernatant was transferred into a 96-well plate (260836, Thermo Fisher Scientific) and mixed with 60 μL water, 80 μL 2× SEAP buffer (20 mM homoarginine, 1 mM $MgCl_2$, 21% (v/v) diethanolamine, pH 9.8) and 20 μL substrate solution consisting of 20 mM pNPP (Acros Organics BVBA). The increase of absorbance at 405 nm in samples was measured using a Tecan Infinite M1000 or Tecan Spark multiplate readers (Tecan AG) for 30 min at 37 °C.

## nLuc measurement

nLuc expression levels were quantified in cell culture supernatants with the Nano-Glo Luciferase Assay System (N1110, Promega). Briefly, 10 μL of buffer/substrate mixture (prepared at a 50:1 ratio) was mixed with 10 μL of the cell culture supernatant in a 384-well plates (781076, Greiner Bio-One). After mixing, the plates were briefly centrifuged at 1200 rpm and incubated for 5 min at room temperature before determining luminescence intensity with a Tecan Spark multiplate reader (Tecan AG).

## GLP-1 quantification

GLP-1 in the cell supernatant was quantified with a High Sensitivity GLP-1 Active ELISA Kit (EZGLPHS-35K, Merck) according to the manual provided by the manufacturer.

## Animal experiments

Animal experiments were performed according to the protocol (Protocol IDs: **20-009-XMQ** and **25-013-XMQ**) approved by the IACUC of Westlake University and in accordance with the Animal Care Guidelines of the Ministry of Science and Technology of the People's Republic of China. Eight-week-old male C57BL/6J and C3H/HeJ mice were ordered through the Laboratory Animal Resources Center of Westlake University. Eight-week-old male BKS.Lepr (db/db) mice were purchased from Shanghai Institutes for Biological Sciences Shanghai Laboratory Animal Center (SLACCAS) (Shanghai, China). Intraperitoneal implants were produced by encapsulating $HEK_{GLP-1}$ cells in alginate-poly-(L-lysine)-alginate beads (about 1000 cells per capsule) using a B-395 Pro Encapsulator (BÜCHI Labortechnik AG; Flawil, Switzerland) set to the following parameters: a 200-μm nozzle with a vibration frequency of 1300 Hz, using 20-mL syringe. Euthanasia was performed by placing the mice in a carbon dioxide ($CO_2$) chamber without prior pre-charging, and 100% $CO_2$ was delivered by displacing 20% of the chamber volume per minute. After 10 min, euthanasia was confirmed by the absence of chest movement, palpable heartbeat, and responsiveness to toe pinch.

## Statistics & reproducibility

Statistical evaluation was conducted by using two-tailed unpaired $t$-test for comparing two sets of data and one-way ANOVA to compare multiple sets of data as implemented in Prism GraphPad 9 (GraphPad Software Inc., San Diego, CA). Unless otherwise stated, all the experiments were conducted with at least three biological replicates, and consistent results were obtained across these replicates. No statistical methods were used to predetermine sample size. Sample size was determined based on similar studies in our lab and other published studies in our field. Attempts at replication were successful. No data were excluded from the analyses. The experiments were not

randomized. The Investigators were not blinded to allocation during experiments and outcome assessment. All samples in this study were allocated randomly. The presentation of data, including sample sizes denoted as biological replicates (*n*), the conducted statistical analyses, and the significance of differences are shown in the figures, and details are provided in the respective figure legends.

## Reporting summary

Further information on research design is available in the Nature Portfolio Reporting Summary linked to this article.

## Data availability

The authors declare that all data supporting the findings of this study are available within the paper, Supplementary Information and the Source Data file. All plasmids used in this study are listed in Supplementary Data 1 and are available upon request from the corresponding author. Source data are provided with this paper.

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

## Acknowledgements
The study was funded by a European Research Council advanced grant (ElectroGene, No. 785800, awarded to M.F.) and in part by the National Centre of Competence in Research (NCCR) for Molecular Systems Engineering.

## Author contributions
N.F., A.P.T., and M.F. designed the project, N.F., S.G., and Z.C. conducted the in vitro experiments, M.X. and S.L. designed and performed the animal experiments, N.F., A.P.T., M.X., and M.F. analysed the data and wrote the manuscript.

## Competing interests
The authors declare no competing interests.
