## [Peer Review File · Nature Communications]

Regulation of therapeutic protein release in response to circadian biomarkers

Corresponding Author: Professor Martin Fussenegger

Version 0:

Reviewer comments:

Reviewer #2

(Remarks to the Author)

The manuscript from Franko and co-authors explores the use of the melatonin receptor 1A (MTNR1A) as a sensor to regulate gene expression in response to the body's natural circadian rhythms. Researchers engineered a melatonin-inducible gene switch that activates transgene expression through the G α s signaling pathway, selectively responding to nighttime melatonin levels. As a proof-of-concept, engineered cells implanted in mice successfully translated circadian signals into regulated GLP-1 secretion at night.

A pro of the manuscript is the engineering of a novel switch that can connect transcriptional control of desired gene of interests (GOI) to the varying levels of melatonin following the circadian rhythms.

There are some points though that require deeper discussion by the authors and that could improve the general interest of such a system.

For example, a con of this work is the coupling to GLP-1 that is easily provided by a pill. But this is rather a conceptual limitation since the GOI can be anything else according to the type of application. Here I'd like the authors to discuss a bit more in detail what kind of potential is there for this technology. There is a mention on the utility in the discussion section but it is rather a list that a conceptualized framework. This can undermine the value of this design that could be perceived rather as a nice exercise of means of transcriptional controllers.

Another major point is the kinetics of the system. The authors observe upregulation of the GOI 12h post melatonin addition/upregulation, which may render problematic whatever desired rapid downstream effect. However the kinetic was studied only using SEAP as reporter. For the GLP-1 induction were quantified 24h to 48h later. Were earlier time points taken? Why this difference in timing?

In the results section more details should be provided, such as experimental set up what was transfected in the cells etc. Some informations are provided in the caption of the figures, but it would be easier for the reader to have it described in the main text.

Specific points:

Fig1b high leakiness in the off state: could the authors comment on what they believe is the source of noise?

Line 142 on the characterization of different promoters in combination for optimal GOI expression capacity: given the relevance that the authors describe in achieving optimised expression, works that focus on the problem of cell capacity and on the characterization of promoters combination and related footprint on the cells should be cited (Di Blasi et al Nat comms 2023; Frei et al Nat Comms 2020; Jones et al Nat Comms 2020).

Line 145 variations in the spacer sequences impact on the reporter performances. This is very interesting, it would be great to explain in the main text the rationale of the spacer sequences and give details on the pVH421 (what spacer sequence did it have)?

Fig3a: it seems that 1nM (1000pM) is the minimal concentration to increase SEAP. Then the inset show concentration at the pM with a slight increase of SEAP. I suppose it's a visual limit of the Y axis in the main graph of 3A. Since the authors state that the peak of Melatonin is 700pM, why not leaving just the inset that is the range of melatonin detected in the plasma?

Line 162 on the fold induction: could they detail the experiment? What concentration of melatonin was given?

Figure 4 is the proof of concept of the melatonin-based switch in vivo.

I feel it misses some important characterization that should have been similar to what was shown in vitro using SEAP as reporter. For example, kinetics of GLP1 release is missing and only the steady state levels are reported. Connection to the physiological circadian rhythm is missing.

it would be worth carrying out these studies having already in place a mouse model for the study.

Fig4b: It is not clear if the experiment was performed already in mice and if the Melatonin was supplemented. Also the Melatonin is 1000uM (meaning 10^9 pM), I wonder why they went out of scale this much.

Fig.4c: the illumination provides very limited reduction of melatonin. I wonder why this was set to be artificial rather than carrying out more physiological night/day monitoring, especially having claimed it can be connected to the circadian clock. Also the authors mention that melatonin profiles and/or sleep behaviours are different between mice and humans. Would be important to discuss that a bit more since this is an in vivo characterization of the system.

Minor: Fig 2 depicts GOI while in the caption they describe POI. Should be consistent with the graphics.

Fig.3 and 4 mention some plasmids name but they do not say what they are.

Caption of Figure 4b describe melatonin 1nM but in the graph it's reported 1000uM which is 10^6 nM.

Fig4. Reports on n=3 independent experiments whereas the previous captions describe biological replicates. I imagine the meaning is the same?

Reviewer #3

(Remarks to the Author)

Overall, this is a methodologically innovative study that developed a gene switch system based on mammalian circadian rhythm regulation to achieve the expression of therapeutic proteins. Some suggestions are listed below to improve readability and clarity.

1. In lines 26-29, more detailed data can be added, such as specificity of circadian secretion and the efficacy data from animal models.
2. In lines 66, cortisol is mentioned, and line 74 could explain why this receptor was chosen instead of the cortisol receptor.
3. When "PCR" and "GOI" are first mentioned, their full names should be provided.
4. It is recommended to supplement the animal experimental data. Lines 230 - 231 indicate that the strategy has the potential to restore normal blood glucose levels in type 2 diabetic mice. If there are relevant animal experimental data, it would be highly beneficial. The addition of such data would greatly enhance the completeness and persuasiveness of the manuscript.

Reviewer #4

(Remarks to the Author)

This study by Franko et al represents an interesting idea of delivery of therapeutic protein in response to circadian biomarkers, such as melatonin. Recently, similar proof of principle clock-controlled genetic delivery circuits in cells have been published, although using a different strategy, i.e., clock gene promoters. As such, the topic is highly topical and novel. The authors nicely illustrated how they identified MTNR1A and its downstream signaling cascade as an effective in vitro system for cell engineering and melatonin-regulated drug/protein release. Extensive constructs have been generated to reach the end goal. In vitro data are largely promising and seem to show a clear melatonin dependent response. However, throughout the paper, there are no statistics (p value), which makes it impossible for the readers and this reviewer to assess its significance. In Data analysis and visualization section, it says: "Statistical evaluation was conducted by using Welch's two-tailed t-test for comparing two sets of data as implemented in Prism GraphPad 9 (GraphPad Software Inc., San Diego, CA)." However, such analysis results were never shown. Besides, it might be appropriate to conduct ANOVA with post hoc tests for some of the time/dose-dependent measurements. In addition, most experimental measurements have very small n numbers, N=2-3.

A major defect of this work is that unfortunately, C57BL/6 mice were used to assess the in vivo effect of illumination on blood melatonin and GLP-1 release from the engineered cells. This in-bred mouse strain is well known to have defective melatonin synthesis pathway (<https://doi.org/10.1073/pnas.091439910>), as such, should be avoided for studies of melatonin. As such, it is unclear how the data in Fig. 4C was generated. The exogenous melatonin treatment experiment is ok in this mouse model. The authors should consider using C3H mice instead for studies of endogenous melatonin.

Another comment is about the ~12-hour delay in SEAP response to melatonin (Fig. 4d). This literally means that in vivo, a sense-response drug would only be released in the day phase following night melatonin exposure. It is unclear how this system could be useful for a timely response to a circadian biomarker. Is this sophisticated system any better than a simple clock gene promoter controlling expression of a gene of interest? I would imagine the latter will likely respond to circadian time cues more quickly without much of a delay. Unfortunately, the paper is extremely short and concise with very little in depth discussion as to how exactly the system will be useful in drug delivery or chronotherapy. Relying on endogenous melatonin would elicit a 12-hour delay, while this reviewer doubts external melatonin could be broadly applied to human subjects.

Version 1:

Reviewer comments:

Reviewer #2

(Remarks to the Author)

The authors have provided satisfactory answers and revision.

I have no further comments

Reviewer #3

(Remarks to the Author)

This study by Franko et al is highly topic and novel. Their careful revision is evident: the manuscript now features numerous new figures and a markedly strengthened mechanistic analysis, bringing it in line with the journal's standards for publication.

Reviewer #4

(Remarks to the Author)

The authors have sufficiently addressed my concerns. The addition of C3H mouse data are particularly welcomed. The added discussions are also appropriate. I support the publication of this exciting work.

Minor comment to the authors, the fact that Melatonin is FDA approved and OTC in the US does not justify its wider applications to the public, as Melatonin is highly regulated in many other countries.

NCOMMS-25-08939 - Response to the reviewers' comments

Reviewer 2:

The manuscript from Franko and co-authors explores the use of the melatonin receptor 1A (MTNR1A) as a sensor to regulate gene expression in response to the body's natural circadian rhythms. Researchers engineered a melatonin-inducible gene switch that activates transgene expression through the G α s signaling pathway, selectively responding to nighttime melatonin levels. As a proof-of-concept, engineered cells implanted in mice successfully translated circadian signals into regulated GLP-1 secretion at night. A pro of the manuscript is the engineering of a novel switch that can connect transcriptional control of desired gene of interests (GOI) to the varying levels of melatonin following the circadian rhythms.

We are grateful for the enthusiastic comments by this reviewer highlighting the novelty of our gene switch controlled by the circadian rhythm.

1. There are some points though that require deeper discussion by the authors and that could improve the general interest of such a system. For example, this work is the coupling to GLP-1 that is easily provided by a pill. But this is rather a conceptual limitation since the GOI can be anything else according to the type of application. Here I'd like the authors to discuss a bit more in detail what kind of potential is there for this technology. There is a mention on the utility in the discussion section but it is rather a list that a conceptualized framework. This can undermine the value of this design that could be perceived rather as a nice exercise of means of transcriptional controllers.

We thank the reviewer for this insightful commentary. Indeed, GLP-1 serves as a model therapeutic protein validating the dynamics of the novel gene switch controlled by the circadian rhythm. We have highlighted this fact in the revised manuscript, expanded on potential general uses of our novel gene switch and compared cell-based vs. pill-based therapies with respect to dosing precision, intervention dynamics, treatment tolerance and resistance. Overall, we believe the discussion section now offers a more integrated perspective on potential gene switch applications in future cell-based therapies. We have also added new *in vivo* data (**new Figs. 4f, 4g**) showing melatonin dose-dependent GLP-1 production and restoration of blood-glucose levels in type-2 diabetic db/db mice (see point 3 of reviewer 3).

2. Another major point is the kinetics of the system. The authors observe upregulation of the GOI 12h post melatonin addition/upregulation, which may render problematic whatever desired rapid downstream effect. However the kinetic was studied only using SEAP as reporter. For the GLP-1 induction were quantified 24h to 48h later. Were earlier time points taken? Why this difference in timing?

As for any transcription-control gene switch, trigger-inducible transgene expression is typically observed as early as 6 h after induction, which reflects the time required for the cells to process product protein production along the transcription-translation-secretion trajectory. The transcription control dynamics of our melatonin gene switch

is similar to that of other gene switches (Foight et al., Nature Biotechnology 37, 1209 – 1216, Liu et al., 2018, Nucleic Acid Research 46, 9864 – 9874, Franko et al., 2021, Nature Communications 12:6786, Ausländer et al., 55, Molecular Cell 397 – 408). Most importantly, the current expression kinetics precisely matches the goal to provide "one dose per day". Nevertheless, we now provide new induction kinetics with improved temporal resolution to capture the onset and dynamics of SEAP and nLuc expression in vitro, as well as GLP-1 secretion in vivo (**new Fig. 4c**).

3. In the results section more details should be provided, such as experimental set up what was transfected in the cells etc. Some information is provided in the caption of the figures, but it would be easier for the reader to have it described in the main text.

We have now included critical experimental details in the results section as well as in figure captions of the revised manuscript.

4. Fig1b high leakiness in the off state: could the authors comment on what they believe is the source of noise?

Since the same reporter construct (a promoter containing cAMP-responsive elements driving SEAP expression, P_{CRE} -SEAP) was used across all conditions (TSHR, MTNR1A, and MTNR1B), any differences in background expression can be attributed to the specific receptor being expressed. To control for this, we repeated the transient transfections side-by-side and included negative controls in which cells were co-transfected with the P_{CRE} -SEAP reporter and a fluorescent protein-expressing plasmid instead of the receptor, maintaining equal total DNA across all conditions. In these controls, basal SEAP levels were low and comparable for both TSH and melatonin inducers. In contrast, cells expressing TSHR showed markedly higher background SEAP expression in the absence of TSH, relative to MTNR1A-expressing cells in the absence of melatonin, indicating receptor-specific basal activity. This is now clearly described in the revised Results section.

5. Line 142 on the characterization of different promoters in combination for optimal GOI expression capacity: given the relevance that the authors describe in achieving optimised expression, works that focus on the problem of cell capacity and on the characterization of promoters combination and related footprint on the cells should be cited (Di Blasi et al Nat comms 2023; Frei et al Nat Comms 2020; Jones et al Nat Comms 2020).

We are grateful for this suggestion and have cited this work in the revised version. In addition, we have found another recent study, which we have also cited (Pferdehirt et al., 2025, Nature Communications 16:1457).

6. Line 145 variations in the spacer sequences impact on the reporter performances. This is very interesting, it would be great to explain in the main text the rational of the spacer sequences and give details on the pVH421 (what spacer sequence did it have)?

We agree and have therefore updated the genotype information on the respective plasmids in revised Table S1, as well as in the main text mentioning these plasmids.

7. Fig3a: it seems that 1nM (1000pM) is the minimal concentration to increase SEAP. Then the inset show concentration at the pM with a slight increase of SEAP. I suppose it's a visual limit of the Y axis in the main graph of 3A. Since the authors state that the peak of Melatonin is 700pM, why not leaving just the inset that is the range of melatonin detected in the plasma?

We are sorry for the confusion associated with the previous version of Figure 3a (now revised as Figure 3b). The figure has been updated to highlight melatonin concentrations that are physiologically relevant.

8. Line 162 on the fold induction: could they detail the experiment? What concentration of melatonin was given?

We have now provided experimental details including the melatonin concentration in the main text and caption of Fig. 3b.

9. Figure 4 is the proof of concept of the melatonin-based switch *in vivo*. I feel it misses some important characterization that should have been similar to what was shown *in vitro* using SEAP as reporter. For example, kinetics of GLP1 release is missing and only the steady state levels are reported. Connection to the physiological circadian rhythm is missing. It would be worth carrying out these studies having already in place a mouse model for the study.

We now provide detailed melatonin induction kinetics of GLP-1 in C57BL/6 mice (**new Fig. 4c**), showing that the gene switch retains comparable activation kinetics to the previously characterized *in vitro* kinetics (**new Fig. 3b**; see also point 3 of reviewer 2 above).

Furthermore, following the advice of reviewer 4, we have turned to C3H/HeJ mice, which may be a more suitable strain to study melatonin-based circadian rhythm-controlled one-dose-per-day transgene expression and to demonstrate the physiological relevance of our gene switch (**new Fig. 4d**). The new results confirm that the melatonin synthesis pathway of C57BL/6J mice could be impaired (**new Fig. S3**; also see specific point no. 8 of this reviewer below), whereas C3H/HeJ mice show circadian rhythm-dependent melatonin production and melatonin-dependent GLP-1 production.

10. Fig4b: It is not clear if the experiment was performed already in mice and if the Melatonin was supplemented. Also the Melatonin is 1000uM (meaning 10^9 pM), I wonder why they went out of scale this much.

We have revised the caption of Fig. 4b to clarify that the data show *in vitro* experiments using encapsulated stably transgenic cells that are later used for implantation into mice. We have also corrected the wrong melatonin concentration. In addition, we have

provided a new data set confirming the dose-responsiveness of the melatonin gene switch. Due to the high sensitivity of the melatonin-responsive gene switch in stably transgenic monoclonal human cells, the system is functional across an extremely wide melatonin concentration range. While the dose-responsiveness within the physiologically relevant melatonin concentration range is most relevant for circadian rhythm-controlled therapeutic transgene expression, higher melatonin levels may be useful for basic *in vitro* research, using other animal models or when there is a need to override circadian control to enable exclusive transgene expression modulation.

11. Fig.4c: the illumination provides very limited reduction of melatonin. I wonder why this was set to be artificial rather than carrying out more physiological night/day monitoring, especially having claimed it can be connected to the circadian clock. Also the authors mention that melatonin profiles and/or sleep behaviours are different between mice and humans. Would be important to discuss that a bit more since this is an *in vivo* characterization of the system.

This issue was also pointed out by another reviewer (see point 3 of reviewer 4 below). To more effectively capture the physiological diurnal variations in melatonin levels, we have followed the advice of all reviewers and conducted the same *in vivo* experiment in C3H mice, which is the strain of choice to study the circadian rhythm (**new Fig. 4d**; see specific point no. 6 raised by this reviewer above). We have also quantified the temporal dynamics of endogenous melatonin production and show that C3H mice have higher and more physiological oscillations of blood melatonin levels over the course of 24 h (**new Fig. S3**). Additionally, we have expanded the discussion on differences in melatonin profiles between humans and mice.

Minor: Fig 2 depicts GOI while in the caption they describe POI. Should be consistent with the graphics.

Thank you. We have corrected this mistake.

Fig.3 and 4 mention some plasmids name but they do not say what they are. Caption of Figure 4b describe melatonin 1nM but in the graph it's reported 1000uM which is 10^6 nM.

Thank you for noting this. We have now added genotype information for all plasmids throughout the revised manuscript and also refer to Table S1 listing all plasmids used and designed during this study. Additionally, we corrected the melatonin concentration in Fig. 4b (please see point on Fig. 4b above).

Fig4. Reports on n=3 independent experiments whereas the previous captions describe biological replicates. I imagine the meaning is the same?

We have corrected this mistake.

Reviewer 3:

Overall, this is a methodologically innovative study that developed a gene switch system based on mammalian circadian rhythm regulation to achieve the expression of therapeutic proteins. Some suggestions are listed below to improve readability and clarity.

We are grateful for these enthusiastic comments.

1. In lines 26-29, more detailed data can be add, such as specificity of circadian secretion and the efficacy data from animal models.

We have revised the introduction to include more detailed information on circadian hormone secretion and associated animal models.

2. In lines 66, cortisol is mentioned, and line 74 could explain why this receptor was chosen instead of the cortisol receptor.

The introduction section was expanded to provide a more thorough explanation of why cortisol has not been chosen as an input for a circadian rhythm-specific gene switch. Specifically, although cortisol exhibits a well-defined circadian rhythm, peaking in the early morning and gradually declining throughout the day, rendering it an intriguing candidate as a circadian trigger for transgene expression, cortisol is deregulated by external stressors, which can lead to unintentional and uncontrolled regulation.

3. When "PCR" and "GOI" are first mentioned, their full names should be provided.

Done as requested.

4. It is recommended to supplement the animal experimental data. Lines 230 - 231 indicate that the strategy has the potential to restore normal blood glucose levels in type 2 diabetic mice. If there are relevant animal experimental data, it would be highly beneficial. The addition of such data would greatly enhance the completeness and persuasiveness of the manuscript.

We had shown exactly this in **Fig. 4g** (old Fig. 4f). db/db mice are type-2 diabetic and serve as an animal model for human obesity-induced diabetes (Bojar et al., 2018, Nature Communications 9:2318, Mansouri et al., 2021, Nature Communications 12:3388, Mahameed et al., 2025, Nature Biomedical Engineering). To clarify this, we have re-named the animal model as "type-2 diabetic" in Fig. 4g. Following the advice of another reviewer (point 2 of reviewer 2 above), we have now added dose-dependence experiments to show that GLP-1 production could be precisely dosed by oral melatonin intake to reach an efficacy window of at least 50 pM in the bloodstream (Shao et al., 2017, Science Translational Medicine, 9, eaal2298) to trigger antidiabetic effects in db/db mice (**revised Fig. 4g**). This is important as the melatonin switch can not only be used to link transgene expression to the circadian rhythm, but could also

take advantage of the clinically licensed over-the-counter sleeping aid compound to drive expression of biopharmaceuticals in a cell-based therapy setting.

Reviewer 4:

This study by Franko et al represents an interesting idea of delivery of therapeutic protein in response to circadian biomarkers, such as melatonin. Recently, similar proof of principle clock-controlled genetic delivery circuits in cells have been published, although using a different strategy, i.e., clock gene promoters. As such, the topic is highly topic and novel. The authors nicely illustrated how they identified MTNR1A and its downstream signaling cascade as an effective in vitro system for cell engineering and melatonin-regulated drug/protein release. Extensive constructs have been generated to reach the end goal. In vitro data are largely promising and seem to show a clear melatonin dependent response.

Thank you for these enthusiastic comments.

1. However, throughout the paper, there are no statistics (p value), which makes it impossible for the readers and this reviewer to assess its significance. In Data analysis and visualization section, it says: “Statistical evaluation was conducted by using Welch’s two-tailed t-test for comparing two sets of data as implemented in Prism GraphPad 9 (GraphPad Software Inc., San Diego, CA).” However, such analysis results were never shown. Besides, it might be appropriate to conduct ANOVA with post hoc tests for some of the time/dose-dependent measurements. In addition, most experimental measurements have very small n numbers, N=2-3.

We now provide the suggested statistics throughout the manuscript based on a minimum of three independent experiments, which is considered standard in the community (Kim et al., 2019, Nature Chemical Biology 15, 1173 – 1182, Galvan et al., 2022, Science Advances 9:2203193, Bertschi et al., 2023, Nature Chemical Biology 19, 767 – 777, Teixeira et al., 2023, Nucleic Acid Research 51, e85, Franko et al., 2024, Cell Discovery 10:9, Yang et al., 2025, Science 387, 74 – 81).

2. A major defect of this work is that unfortunately, C57BL/6 mice were used to assess the in vivo effect of illumination on blood melatonin and GLP-1 release from the engineered cells. This in-bred mouse stain is well known to have defective melatonin synthesis pathway (<https://doi.org/10.1073/pnas.091439910>), as such, should be avoided for studies of melatonin. As such, it is unclear how the data in Fig. 4C was generated. The exogenous melatonin treatment experiment is ok in this mouse model. The authors should consider using C3H mice instead for studies of endogenous melatonin.

We were indeed struggling with the choice of our animal models when profiling native melatonin-based circadian control. Thus, we are extremely grateful for the recommendation by this reviewer to use C3H mice for circadian rhythm-related experiments. We have followed this advice and show that C3H/HeJ mice were indeed more sensitive to light-dependent manipulation of endogenous melatonin production (new Fig. 4d) and that C57BL/6J mice do have diminished melatonin production

capacity (**new Fig. S3**). Consistently, baseline GLP-1 expression in melatonin-deficient C57BL/6J mice remained below the therapeutically active threshold of ~50 pM (Shao et al., 2017, *Science Translational Medicine*, 9, eaal2298), suggesting that the gene switch may stably remain in an inactive state *in vivo* unless exogenous melatonin is provided. Therefore, C57BL/6J (and also db/db mice) may be appropriate models to describe therapeutic scenarios where defined doses of exogenous melatonin would be used stimulate transgene expression *in vivo* (**Fig. 4e-4g**). In contrast, C3H/HeJ mice may be preferred to study putative circadian rhythm-dependent dosing regimens (**Fig. 4d**). This substantially expands the potential applications of our system, as the melatonin switch can not only be used to link transgene expression to the circadian rhythm (**Fig. 4d**), but could also take advantage of the clinically licensed over-the-counter sleeping aid compound to drive expression of biopharmaceuticals in a cell-based therapy setting (**Fig. 4e-4g**). We have accordingly revised the corresponding sections in the manuscript.

3. Another comment is about the ~12-hour delay in SEAP response to melatonin (Fig. 4d). This literally means that *in vivo*, a sense-response drug would only be released in the day phase following night melatonin exposure. It is unclear how this system could be useful for a timely response to a circadian biomarker. Is this sophisticated system any better than a simple clock gene promoter controlling expression of a gene of interest? I would imagine the latter will likely respond to circadian time cues more quickly without much of a delay. Unfortunately, the paper is extremely short and concise with very little in depth discussion as to how exactly the system will be useful in drug delivery or chronotherapy. Relying on endogenous melatonin would elicit a 12-hour delay, while this reviewer doubt external melatonin could be broadly applied to human subjects.

It is not clear what this reviewer is referring to. Fig. 4d shows melatonin-tunable GLP-1 expression in mice and not SEAP expression kinetics. In fact, none of the panels in Fig. 4 shows SEAP expression. We therefore assume that this reviewer is either referring to Fig. 3c, showing melatonin-adjustable SEAP expression kinetics or Fig. 3d, demonstrating reversibility of melatonin-triggered SEAP expression. In both cases, melatonin-responsive SEAP expression can be measured a few hours after induction, which is precisely in line with other transcription-control modalities (Foight et al., *Nature Biotechnology* 37, 1209 – 1216, Liu et al., 2018, *Nucleic Acid Research* 46, 9864 – 9874, Franko et al., 2021, *Nature Communications* 12:6786, Ausländer et al., 55, *Molecular Cell* 397 – 408). The apparent delay in ramping up protein expression results from a combination of reporter protein assay sensitivity and the need for the reporter proteins to be processed along the transcription-translation-secretion trajectory, which produces a delay inherent to all transcription-control elements. In fact, a recently reported clock promoter (Pferdehirt et al., 2025, *Nature Communications* 16:1457) shows comparable induction kinetics, compatible with the intended one-dose-per-day regime, though the melatonin system offers higher maximum expression levels and lower leakiness. We have revised the discussion part to better put the melatonin switch into perspective as (i) a biocompatible gene switch using a physiological compound for *in vitro* gene-function analyses, (ii) a potential one-dose-per-day biopharmaceutical production and release system in future cell-based therapies when connected to the melatonin-based day and night cycle and (iii) a tool for melatonin-triggered transgene expression in future gene and cell-based therapies. Since melatonin is an FDA-licensed over-the-counter sleeping aid taken by millions of

people every day, the combination of combining sleep with biopharmaceutical in-situ production and release is a welcome prospect, in particular for elderly patients. In this context it remains unclear why this reviewer is questioning the broad application of external melatonin in human subjects, as it is already broadly applied by self-dosing, without any known side effects such as dependence, tolerance or addiction (Andersen et al., 2016, Clinical Drug Investigation 36, 169 - 175).